# Geological Materials in Late Antique Archaeology: The Lithic Lectern Throne of the Christian Syrian Churches

**Giovanna Bucci**

Cultural Heritage Department, University of Padova, 35139 Padova, Italy; giovanna.bucci@unipd.it

**Abstract:** The geological materials used in early Christian Syrian churches involve a lithic furnishing element: the lectern throne of the Syriac bema, a stone device used as a support for the holy books. Some inscriptions found in Syria suggest an interpretation for this artifact, located in the middle of the Syriac bema hemicycle, fronting the altar zone. These elements were made of basalt or limestone, depending on the geographical–geological context of the building. In this work, an unedited classification of the main typologies of thrones is proposed with a *collatio* between geo-archaeological data, epigraphic texts, mosaic inscriptions, literary sources, and findings. The role of this uncommon piece of furniture, uncertain up to now, is explained with a new interpretation coming from archaeological–architectural data combined with ancient sources. The study thus locates this architectonical sculpture in the building stratigraphy and also describes decorations from the lecterns, thus contributing to chronology analysis of published and unedited Syrian sites.

**Keywords:** lectern throne; Syriac bema; church; Christian archaeology; Syria

## 1. Introduction

Geological materials in Syrian late Roman architecture are an important component of cultural heritage, especially in the ambit of Christian archaeology. The stones were used to build churches as well as elements of their interior design, including religious objects and furnishings, like the lectern throne.

Lithic heritage from northern Syria is a source of knowledge for Christianity studies, holding persistent cultural information recorded in geological materials and supported by inscriptions.

Thanks to the characteristics of the geological materials chosen during Late Antiquity by the architects, we have a great collection of structures still surviving in situ. The visual impact created by massive buildings is a demonstration of a vivid idea of communication of religious themes.

Christian Syrian churches between the 4th and the 6th centuries AD were built with the geological material available on-site: limestone and basalt. Local natural resources were preferred, except for some special liturgical furnishings for the altar and the ciborium (made of precious smoothed marbles from across the Mediterranean area). The geomorphological condition was decisive for the choice of building materials. The possibility of extracting lithic material directly on-site enabled a great perspective in economic and energy-saving terms and allowed good agility for the work in progress. The Limestone Massif and the Basaltic Plateau from the late Pliocene are the geological units that were used to quarry for constructions because of their natural stratigraphy characterized by horizontal layers, which made them easy to cut and use as building materials [1–4]. This architecture is in fact characterized by monolithic blocs in the construction of walls, as well as in covering systems, floors, windows, doors, and architectonical elements, and has capitals that show very homogeneous massive structures [5].

## 2. Architectonical Characteristics: Buildings and Thrones

The buildings under study, widely diffused in the region of the Limestone Massif, but also near Hama and Idleb (Figure 1), are characterized by longitudinal development

and they are usually divided into three naves by rows of columns; the presbytery area is organized with a *cancellum*, *synthronon*, and altar. In front of the holy structure, the bema, there is also the Syriac bema, a large horseshoe-shaped platform located on the main axe of the central nave of the church. The hemicycle is flanked by sitting banks and joined with the lectern throne, a lithic or a wooden liturgical furnishing element (Figure 2; for more information about the bema churches, see [6–9]; for a definition of Syriac bemas, see [10–14]; for information about bemas and the connection with Syriac bemas, some literary sources, and a possible reconstruction of the paraments, see [15–18]).

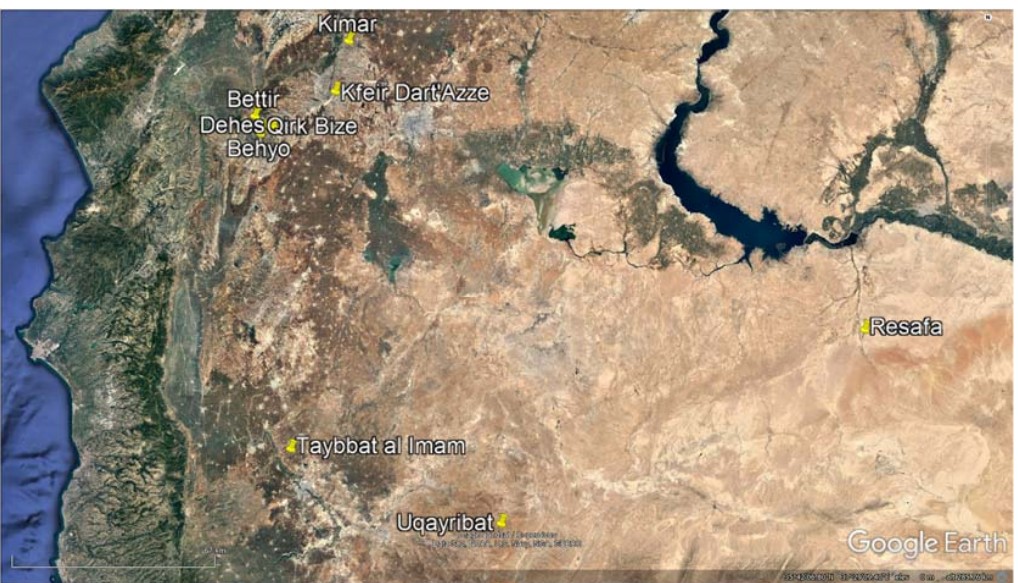

**Figure 1.** Google Earth satellite image of north and central Syria with the archaeological sites mentioned in this work (accessed 23 December 2020).

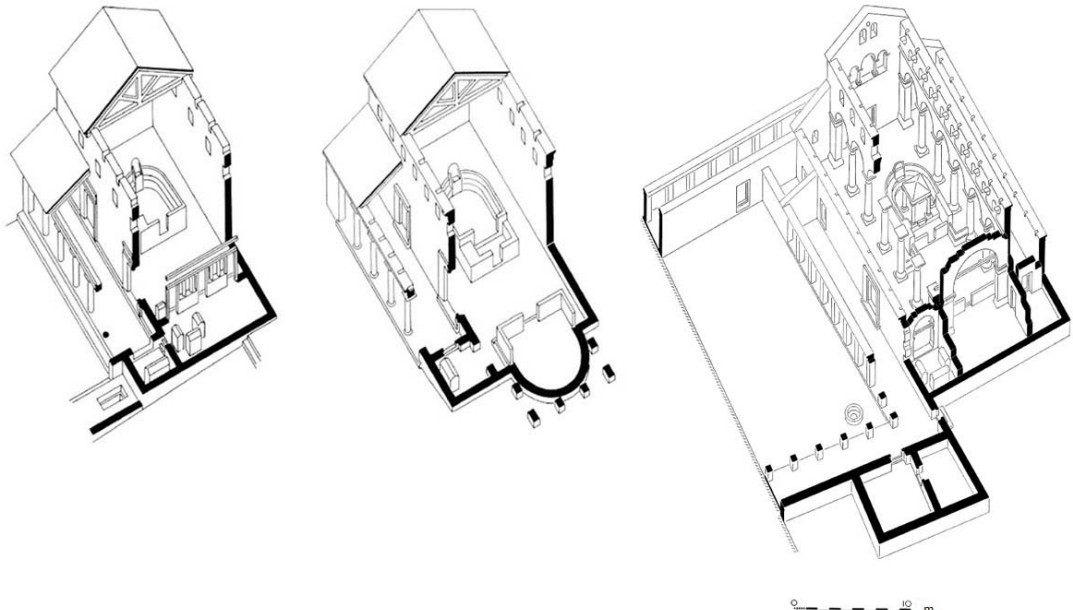

**Figure 2.** Qirk Bize, Kfeir Daret 'Azze, and Behyo churches: axonometric reconstruction (Tchalenko 1990).

Despite there being more than fifty Syriac bema churches [10,15], only nine exemplars of lectern thrones have survived; they come from Kfeir Dart 'Azze, Kimar, Bafetin, Dehes,

Resafa, Qirq Bize, Behyo, Bettir, and Bennawi. Except for the last one, which is on display in the garden of the Damascus National Museum, they are all on site.

The surviving artifacts are made of calcar, limestone, or basalt, with a quite trapezoidal or L-shaped section and a parallelepiped base. From the morphological point of view, it is possible to elaborate a classification of three main typologies: (a) a monolithic block with tilted book support and acroteria globes, with a trapezoidal section (Bennawi, Qirk Bize, and Resafa); (b) a monolithic block with a simple L-shaped section (Kimar); (c) a composition of lithic elements with a rounded dossal (Kfeir Dart 'Azze, Bafetin, Dehes, Behyo, and Bettir) (Figure 3).

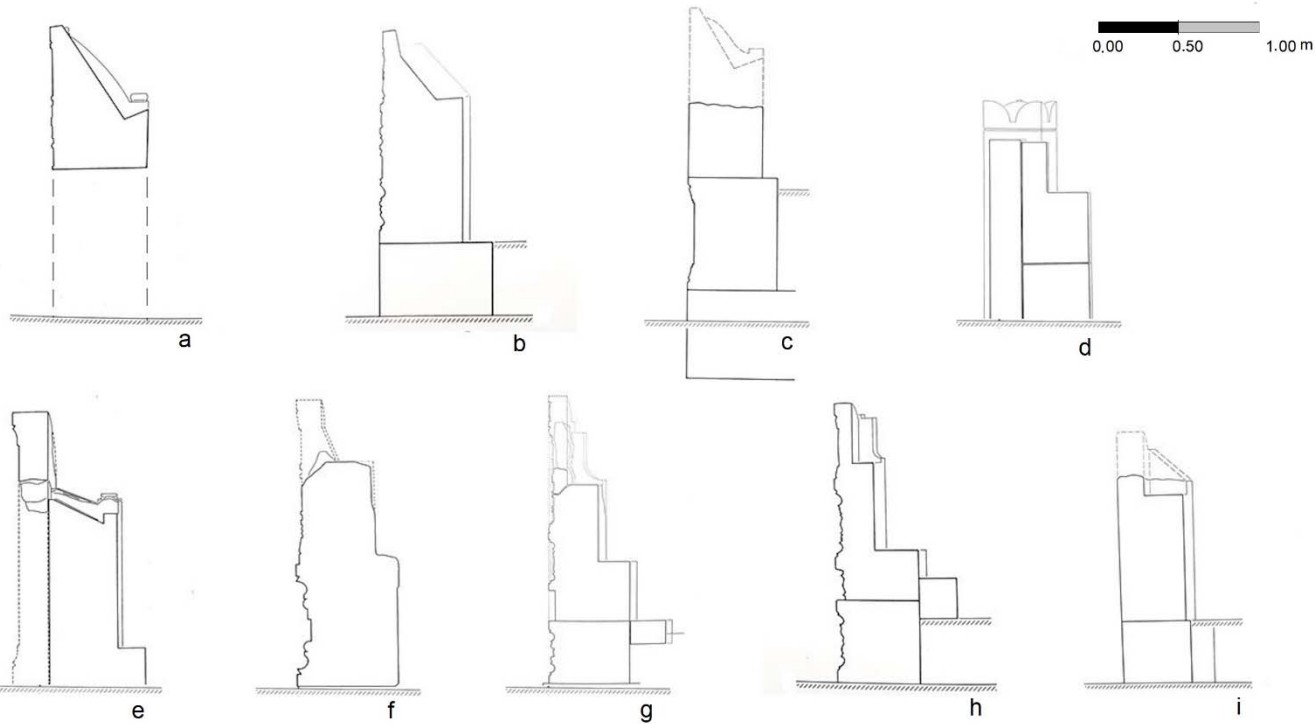

**Figure 3.** Profiles of the preserved lectern thrones: (**a**) Bennawi, (**b**) Qirk Bize, (**c**) Resafa, (**d**) Kimar, (**e**) Kfeir Daret 'Azze, (**f**) Bafetin, (**g**) Dehes, (**h**) Behyo, and (**i**) Bettir (elaboration of tables from Tchalenko 1990).

The lectern thrones were very dense and massive artifacts; only in the upper parts do some architectural details lighten this liturgical furniture. They would have been standing on stone foundations, decorated to the rear with geometrical patterns, like circles, crosses, meanders of swastikas, and phytomorfic rinceau (Figure 4), as well as inscriptions (see *infra*).

The thrones were probably built to serve a special session of the celebration, during which the clergy would move from the presbytery to the center of the main nave to read the holy texts (important texts of worship and prayer, as explained below). The books were placed on the slightly sloped support facing the altar zone on the lectern throne. The purpose of the Syriac bema was perhaps to enable the weekly reenactment of the crucifixion and resurrection through its use in the liturgy of the Word. When it was time for the reading of the Gospel, the clergy would leave the sanctuary and carry the Word. From the symbolic point of view, the device underlines the connection between the heavenly Jerusalem and the terrestrial Jerusalem, or Golgotha. The complex of the bema with the Syriac bema represents a microcosm inside the church, as the earlier Christian sources explain, perhaps referring also to *hetoimasia*, also connected to Jerusalem and Bethlehem, Paradise and Earth [14,15].

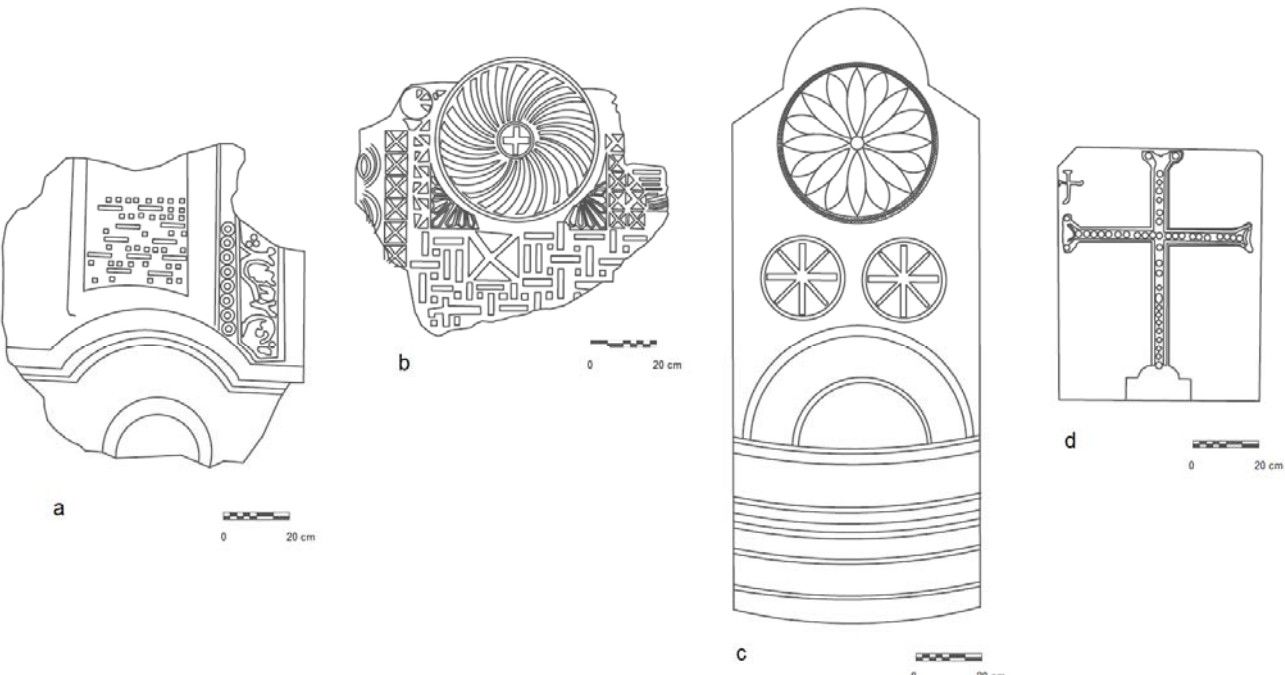

**Figure 4.** Preserved decorations from lectern thrones: (**a**) Dehes, (**b**) Kfeir Daret 'Azze, (**c**) Qirk Bize, and (**d**) Kimar (elaboration from G. Bucci 2020).

## 3. Philological and Epigraphic Investigations

### 3.1. The Literary Sources

The throne was the most important part of the Syriac bema. Christian literary sources underline the presence of a small table in front of the throne, a sort of small, sigma-shaped altar, complementary to the throne. Examining the *Patrologia Graeca* and *Scriptores Syri*, we found texts explaining celebration movements and prayers that document the presence of the Syriac bema with its peculiar structure (Farioli, Zaqzuq, and Piccirillo suggest the text found in the *Historia Ecclesiastica* by Eusebius VII, 30, 9; here, the terms "bema" and "throne" are used together). Eusebius mentions a letter written by the bishops in the Synod of Antioch against Paul from Samosata: "It is not even our duty to examine the ambitious vanity of whom, in the church's assemblies, aimed at nothing else but vain glory and pomp, at impressing inexpert souls with such shrewd, who has had prepared for himself a court and raised throne, the like of which a disciple of Christ should not have".

Remarkable considerations from the *Corpus Scriptorum Christianorum Orientalium* have been exposed by Sozomen in *Historia Ecclesiastica*, VIII, V, p. 1528–Migne PG 67. Sozomen explains the position of the Syriac bema with its throne; other specifications are provided by Connolly in the 1913 edition of the *Anonymi Auctoris Expositio Officiorum Ecclesiae Georgio Arbelensi Vulgo Adscripta*, in the commentary about the *Tractatus* II.11. Connolly underlines that *Bema in medio templo positum est.* The Tractatus IV, XIII–XV, gives other details: *Elevatum fuisse bema a templo certum est; namque ministri illum semper ascendere dicuntur. Spatiosum quoque fuisse necesse est, quia episcopus ibi, una cum presbyteris assistentitibus. Manebat usque ad finem reponsoruu mysteriorum, quod cantabatus post missionem catechumenorum.*

More data come from Gabriel bar Lipeh of Qatar in his *Commentary to the Liturgy*, 9 and 16, where we find some aspects concerning the moving of the clergy during prayers and readings [19–23].

The *Glossarium ad Scriptores Mediae et Infimae Graecitatis*, sub vocem βῆμα (bema) gives a general definition of the presbytery area, Locus in Templo ubi consistent Sacerdotes, mentioning Canonis 69 of the Synodo Trullana, the Mystagogy of Germanus, the Carmina of Gregorius Nazianzienus, Palladius, Eustatius, and Simeon Tessalonicensis; a very important explication is reported in the lemma βηματικά (bematica), *preces que in Ambone, aut in*

*solea, à Cantoribus vel Psaltis in Ecclesia concinunutur*, also referring to the Psalmi graduales, whose reading or intonation could have been carried out near the bema, the ambo, or perhaps in the Syriac bema. About the word "θρόνος" (thronos) [24], the Glossarium mentions a "Θρόνος δεύτερος" (thronos deuteros), a place to reach during liturgy, outside the presbytery area. Maybe the βηματικά prayers were performed in the Syriac bema using the lectern throne, the θρόνος δεύτερος (thronos deuteros), as support for the holy texts. This means that the lectern throne was dedicated to these special readings.

### 3.2. Epigraphical Sources

From the epigraphical perspective, there are some remarkable indications concerning the structure and the position of the lectern throne: the documentation comes from Greek, Syriac, and Judaic inscriptions.

A Princeton University expedition, coordinated by Butler in 1899, documented in Zebed a Greco-Syriac inscription on a slab of the *cancellum* of the presbytery area mentioning a *thronos*: *AR(D)A Rabūlā made the throne* (for more information about Zebed, see the Archaeological Archives at http://vrc.princeton.edu/archives/items/show/47085) [25]. It is an attestation of the erection of a throne by Rabula. The literal translation is "made the throne", but that, of course, does not necessarily mean that he made it himself. In this case, undoubtedly, it is the acknowledgment of a gift [26] (Figure 5).

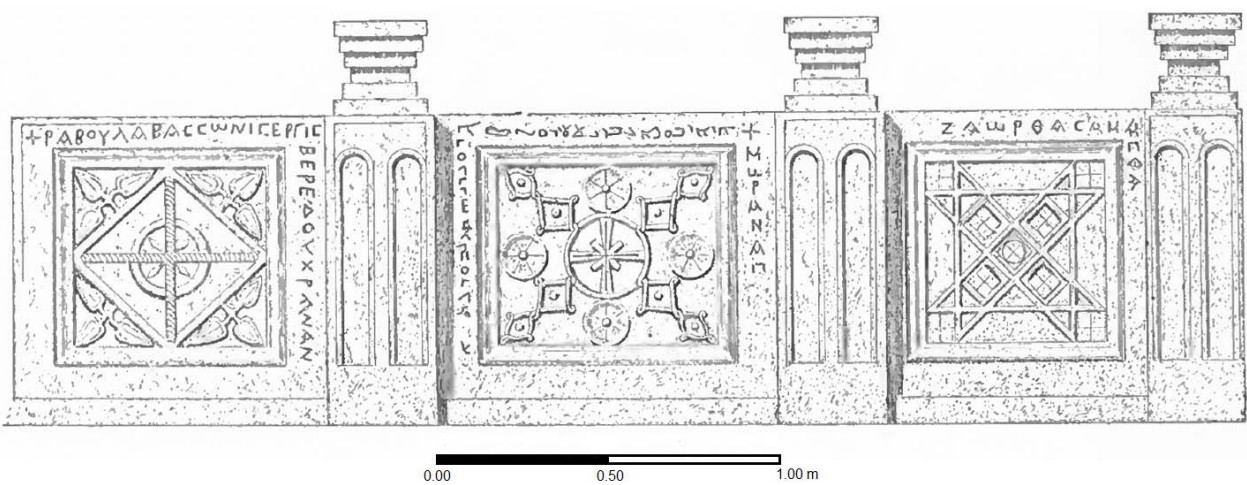

**Figure 5.** Zebed: cancellum of the bema with inscription (Littman 1930).

The word *thronos* could also be used in a metonymical way, a part for all, underlying, in this Christian archaeological ambit, the use of *thronos* as *vox media*: the U-shaped podium/place and the lectern (the Syriac word translated from *thronos* is sometimes metaphorically used for "altar", as the throne of God). With regard to the donors of the bema and throne, there are also Judaic inscriptions from the first half of the 5th century AD preserved in private collections, published by Salame-Sarkis in 1989, with the following content: *In pious remembrance of Him, whose name is blessed/forever, the good and merciful One/For a throne (?) of the holy god/was this made by bā [ . . . ]*.

A unique case of a bema throne inscription, supported by the artifact itself, is the Bennawi throne, decorated at the rear with a Syriac Greek dedication; the second hypothesis of translation suggested by Chabot explains the direct allusion to the lectern or to the bema throne complex: *For a good remembrance either for the priest Abraham, and for John, and for his mother who had it done* [27] (Figure 6).

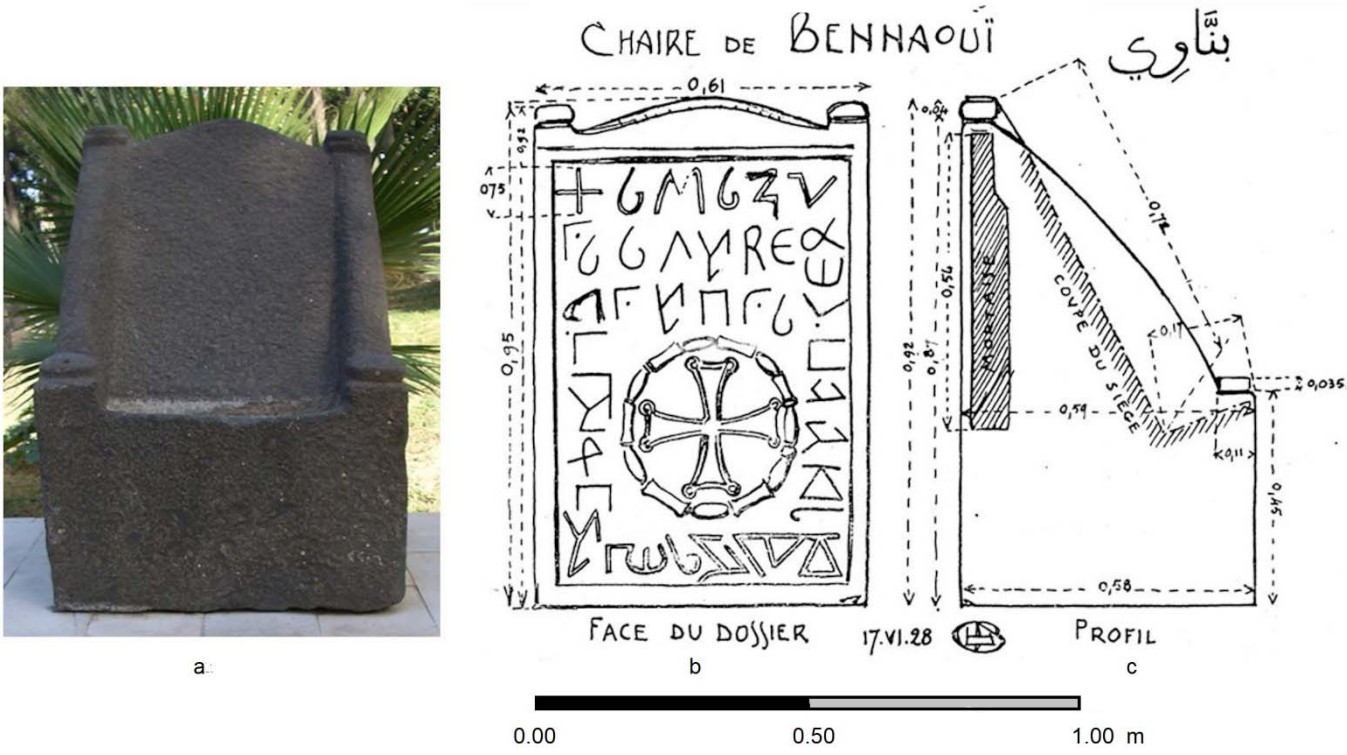

**Figure 6.** Bennawi lectern throne: (**a**) photo by Bucci, (**b**,**c**) drawings by Chabot (1929).

Two more epigraphic documents record interesting data.

The mosaic floor inscription belonging to the second big panel of the central nave of Taybbat al Imam church (442–447 AD), near Hama [23,28,29], mentions the construction of the *thronos* (word used to indicate the throne itself, but also the bema in a metonymical way): *of Thedose praying God with (his) wife and sons paved with mosaic (the area) behind the throne* (Figure 7).

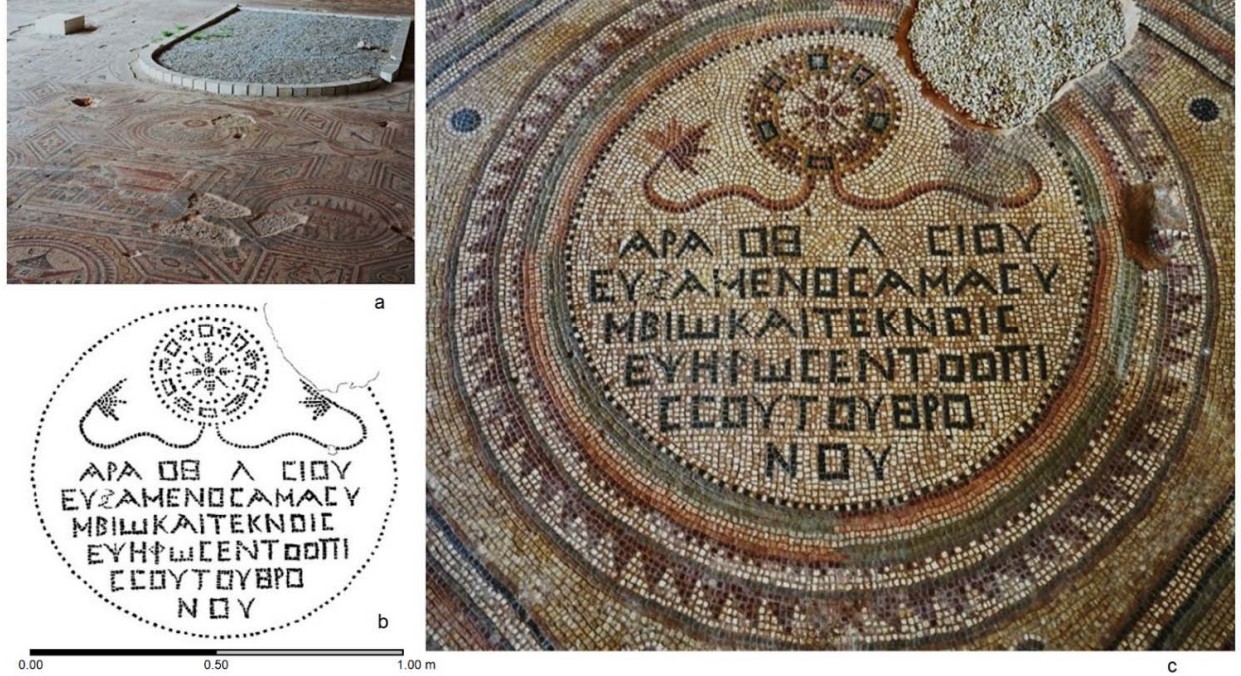

**Figure 7.** Taybbat al Imam: (**a**) perspective image of the Syriac bema, (**b**) relief of the bema throne inscription, (**c**) mosaic with bema throne inscription (Piccirillo, Zaqzuq 1990).

The inscription is immediately west of the Syriac bema, of which only the morphological trace of the foundation remains, thanks to the archaeological excavations and evidenced due to the musealization of the site. Similar U shapes in mosaic floors, fronting the presbytery area, are also attested, for example, in Rayan, at the same phase of the mosaic pavement; moreover, in Banassara, as reported by Khoury in a 2002–2004 excavations report, a lithic structure was over the mosaic floor: it seems that the Syriac bema was added in a second phase of use of the church, after a restoration, covering part of the mosaic decoration [30] (usually, Syriac bemas are instead part of the original project of buildings; see *infra*).

## 4. The Surviving Exemplars of Lectern Thrones: Description

The mentioned Bennawi throne, preserved at the National Museum in Damascus, is the most famous bema throne (see Figure 6); it is made of basalt and comes from the church of Bennawi, south of Aleppo; it is mentioned in the reportage of Tchalenko (1950–1953) and discussed in published work by Chabot [6,17,27]. The preserved section is the upper side, the lectern itself; it is characterized by a vertical dossal terminating with a tympanum on the external part; the oblique frontal part is the support for the book; two pairs of globes, flattened at the ends of the extremities, complete the furniture (0.2 × 0.61 × 0.50 m). The dossal is decorated with the mentioned Syriac inscription on four horizontal lines and two vertical rows: in the middle, a pseudo-laurel clypeus encloses a cross with expanded extremities. At the sides, there are two rectangular housings for the joints with the balustrade of the U complex. The lectern was made between the 5th and the 6th centuries AD.

In Jebel el Ala, the church of Qirq Bize includes a compact limestone structure (Figure 8) holding the Syriac bema, enriched with a monolithic lectern throne, with great external decorations (IFPO expedition: https://medihal.archives-ouvertes.fr/hal-025625 19/thumb) [31]: grooves, inverted grooves, and linear frames with a big arch in the lower part; a big *clypeus* with petals and a small concentric circle is arranged over two twin *clypea* with schematic monograms. The support for the book is inclined in the same way as the Bennawi example. Here, the parapets of the hemicycle are still standing, flanking the lectern. It is attributable to the first half of the 5th century AD (Figure 9).

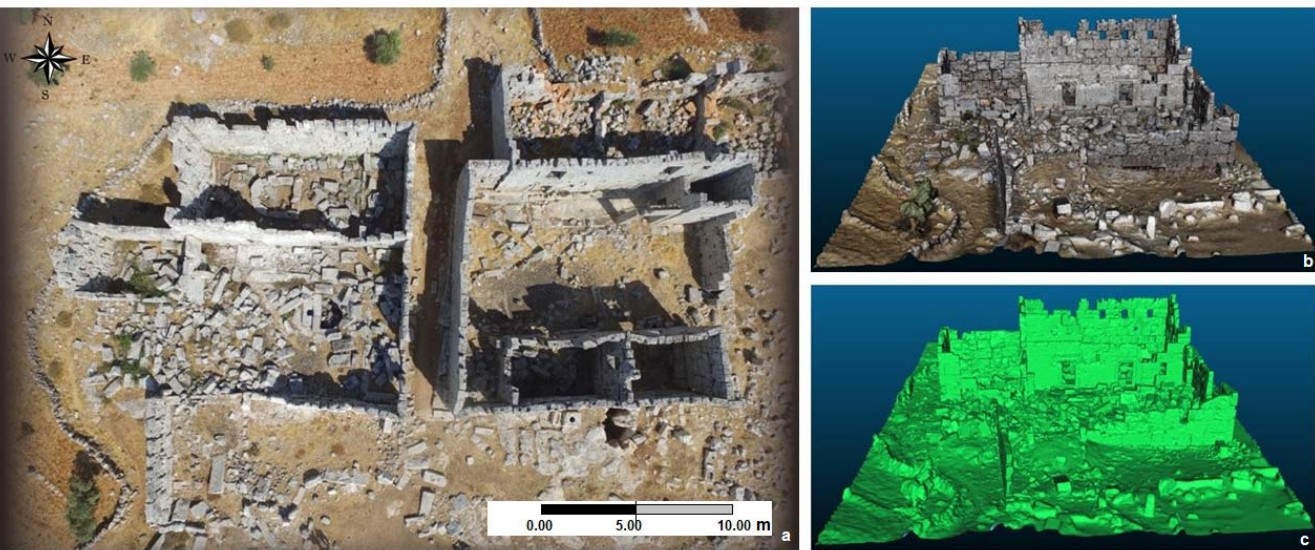

**Figure 8.** Qirk Bize: (**a**) drone image, (**b**) textured 3D model, (**c**) shaded 3D view from the south side (Tsuneki, Watanabe, and Jammo 2020).

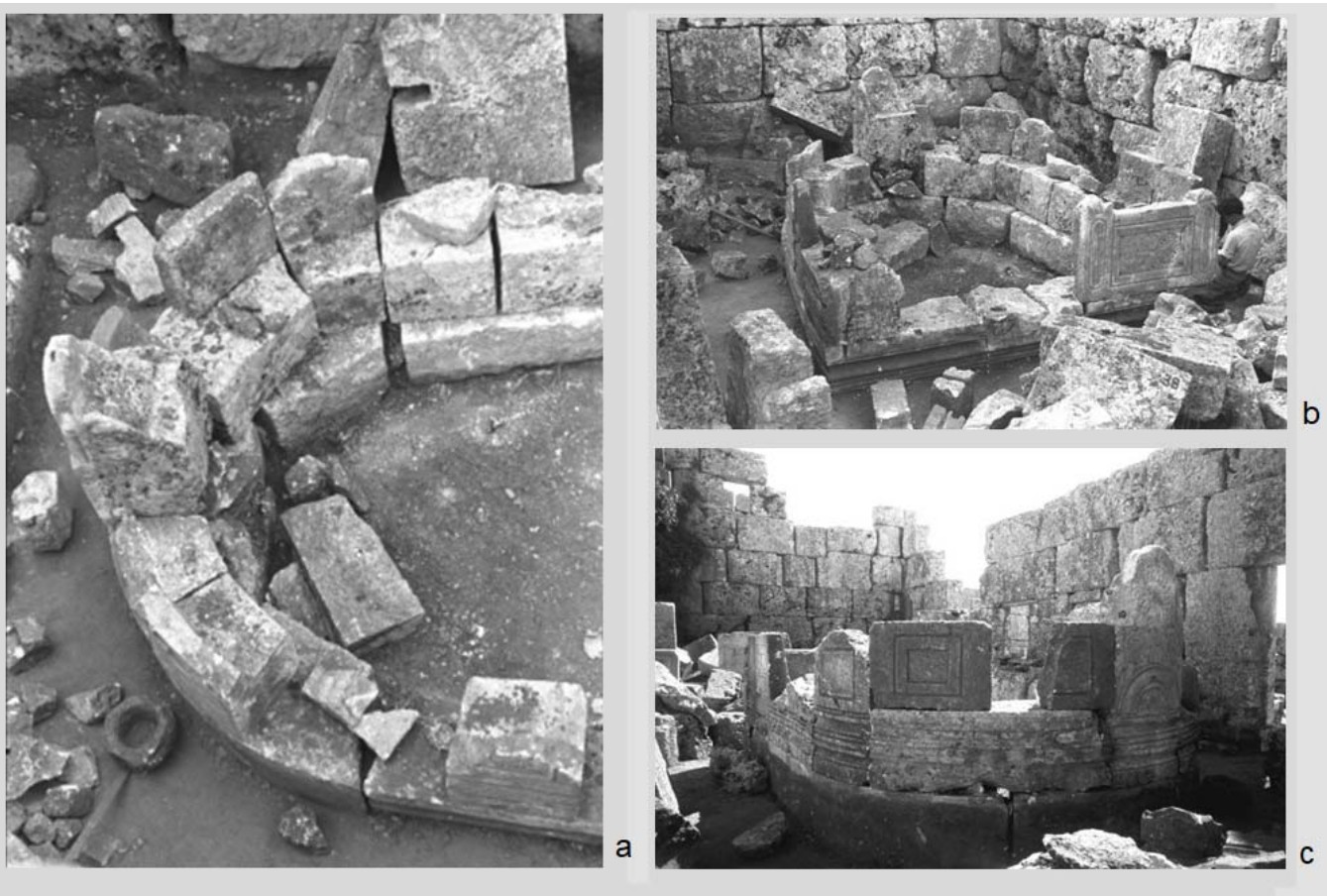

**Figure 9.** Qirk Bize, Syriac Bema with lectern throne: (**a**) U-shaped structure with throne, viewed from above and (**b**) in perspective; (**c**) external part with decorations (elaboration from https://medihal.archives-ouvertes.fr/medihal-01810641/thumb).

Another context with good conditions of conservation is the Holy Cross Basilica in Resafa [32], where the bema platform made of stones is preserved with the steps and the floor in its second phase (second half of the 6th century AD). The throne was made of three big blocks, following Tchalenko's hypothetical reconstruction, and the higher section was similar to the Bennawi lectern (in this case, the lectern was made of limestone).

In the Jebel Seman, in Kimar church (1960 IFPO expedition: https://medihal.archives-ouvertes.fr/search/index/?q=kimar&submit=&submitType_s=file), the lectern was made of complementary limestone blocks (probably four): a monolithic L-section element, decorated at the rear by a cross with gems and expanded extremities, was supported by a simple parallelepiped stone, flanked by a couple of supports with crowned acroteria. The artifact belongs to the first half of the 5th century AD (Figure 10).

In Kfeir Dart'Azze (Jebel Seman), there is a bema church with foundations still standing. In between the ruins, collapsed and partially destroyed, the French mission of the *Institut français du Proche-Orient (IFPO)* detected pieces of the lectern throne, as documented by Tchalenko (1964 IFPO expedition: https://medihal.archives-ouvertes.fr/search/index/?q=Kfeir+Dart%E2%80%98Azze+&submit=&submitType_s%5B%5D=). The furniture was made of two limestone blocks: a vertical dossal with a curvilinear upper section decorated, on the external part, with a clypeus, grids of squares divided by diagonals, a swastika meander, and frames with organic patterns. It is datable to the end of the 4th or the beginning of the 5th centuries AD (see Figure 4). The same type of lectern, from the first half of the 5th century AD, can be found in Bafetin church [1].

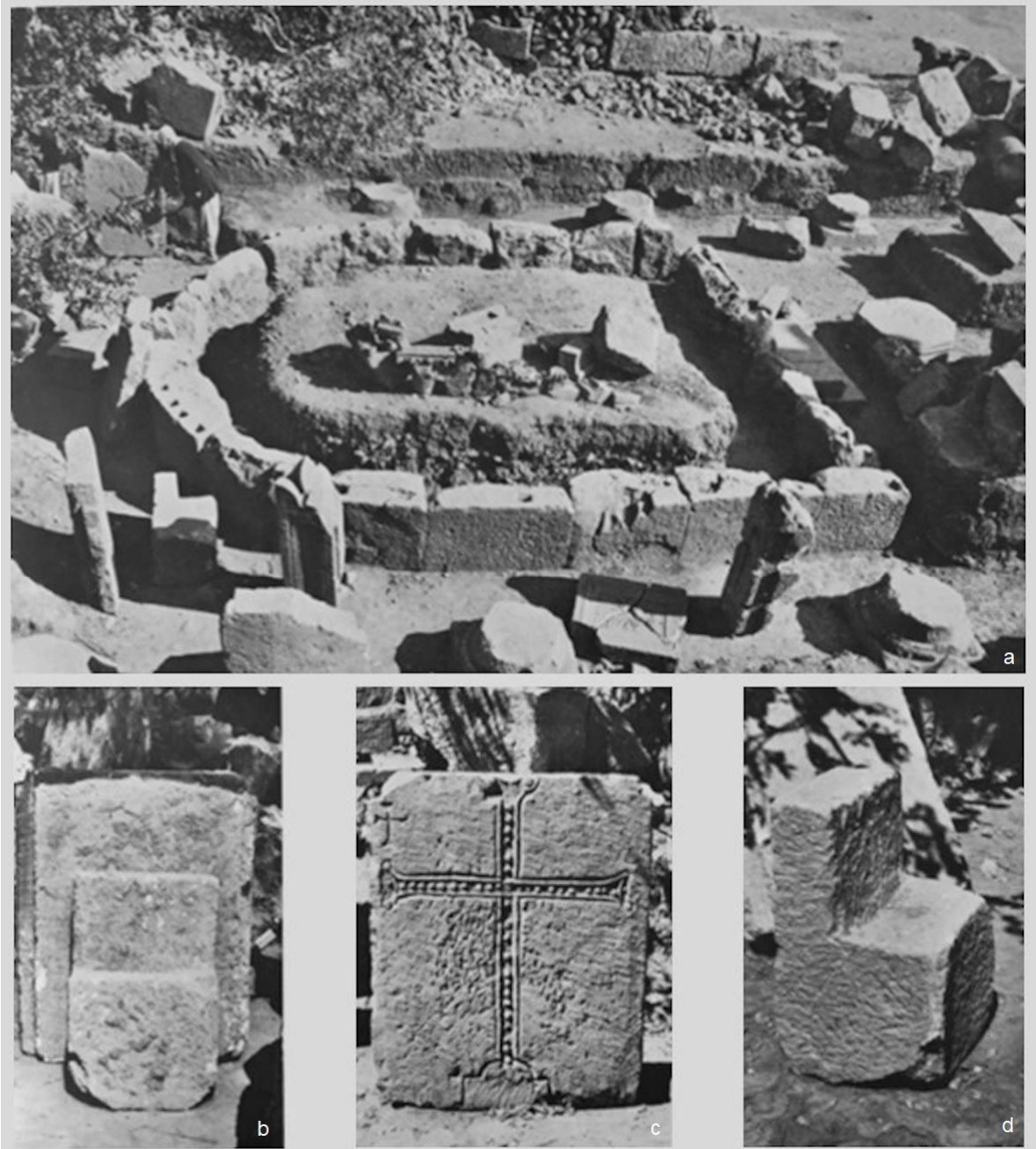

**Figure 10.** Kimar: (**a**) Syriac bema, (**b**) lectern throne internal side, (**c**) external decoration with cross, (**d**) perspective (Tchalenko 1980).

In Déhès basilica (Jebel Barisha), the structure of the Syriac bema is quite well pre-served, including the lectern throne on-site (IFPO expedition: https://medihal.archives-ouvertes.fr/search/index/?q=D%C3%A9h%C3%A8s&submit=&submitType_s=file). The U-shaped wall stands with two coursings of limestone blocks, externally decorated by linear frames, with grooves and inverted grooves. In an emphatic position, the throne shows the continuity in decorations from the same archaeological phase (second half of the 5th century AD); the lower part includes an arch and the central part displays a sort of meander decoration, with two bands with phytomorphic motifs flanking the central panel.

Also in Behyo (Jebel el Ala), there is a small basilica preserving a lectern throne made of three limestone blocks: an upper part, the support, and a mini-platform (IFPO expedition: https://medihal.archives-ouvertes.fr/search/index/?q=Behyo&submit=&submitType_s=file

and Archaeological Archives: http://vrc.princeton.edu/archives/items/show/46562). The dossal is vertical, curvilinear in the upper portion, and decorated on the outside by overlapping linear frames with grooves and inverted grooves; the inside part is structured as three large steps. The artifact is datable to roughly the second half of the 5th century AD. An example made of two blocks was documented by Tchalenko at the site of Bettir (1939 IFPO expedition: https://medihal.archives-ouvertes.fr/search/index/?q=bettir&submit=&submitType_s=file), Jebel el Ala, which is maybe similar to the Kimar throne and attributable to the second half of the 5th century AD.

## 5. Comparisons and Stratigraphical Considerations: The Role of the Stones

Analyzing the relationships between the wall stratigraphy of the Syriac bemas, the floors, and the walls, it is possible to deduct that the lectern thrones were parts of the original projects of the churches. The matrix of the stratigraphic units and the decorative continuity in the alignment of the frames, with the same base elevation and total uniformity of style, demonstrate that the levels of the lecterns on-site belong to the main chronological phase and document a unitary project. In the preserved thrones, we always find the use of the same geological material as the church structure itself. There are the same building methods and techniques, with the same laying spaces and absence of binder in the main walls, as well as in the complex of the bema throne. The installation of the lectern is an integral part of a special architectonical module. The reliefs of the wall structures can confirm the chronological attributions of the artifacts and can also be primary sources of information for the study of the structural design and construction techniques and for dating architectonical sculptures.

Wooden thrones were perhaps present too, but they were supported by lithic elements of local stone; the Syriac bemas, even when stripped of the constructive elements—and therefore to be read through negative stratigraphic units—confirm the typology of the liturgical furnishing project. Two cases are emblematic for the definition of the structures and the periodization of the buildings, as well as for the chronology: Uqayribat and Taybbat.

A new case of a Syriac bema with a U-shaped platform, connected with the mosaic floor, is in Uqayribat, an archeological site in the Hama district, accidentally discovered in 2018: it is a big church with a great mosaic floor, including a U-shaped bema foundation that has been spoiled, preserving the pavement. The site was discovered while the Syrian Army was combing the area for mines. The Directorate General for Antiquities and Museums of Syria (DGAM) worked to uncover the mosaic, which was later removed and transferred to the Hama National Museum under the coordination of the DGAM director Dr. Mahmoud Hamoud (http://www.dgam.gov.sy/index.php?d=177&id=2390). The decorations on the Syriac bema show a panel with a palm tree representing the tree of life, with three couples of heraldic lambs framed by rhombs and triangles tangential to the *tabula ansata*, over which there is a unicorn with two laurel *clypea*. In the middle, the inscription mentions Bishop Alexander, the *indictio*, and the month of the execution of the mosaic. The ditch derived from the masonry stripping (see *infra*) testifies to the presence of a liturgical installation with typical morphology (Figure 11).

If we analyze the context of Uqayribat, reading the layers of the archaeological excavations where it is still possible, we can see, on the basilica floor mosaic, the level of the spoliation trench of the Syriac bema hemicycle lithic structure (the spoliation is in phase with the spoliation of the architectonical elements, including the columns bases, which have for sure been reused somewhere else). The foundation system of the liturgical furnishing, holding all the blocks at the same level, dislocated in an artificial ditch with a flat bed and vertical sides, cuts the leveled splitting plane of the floor; it is possible to see the U-shaped spoliation trench, here emphasized by the 3D plan view and section (Figure 12). The continuity of decorations holding the U structure, including the throne, confirms the shape of the architectural project. The mosaic was completed after the construction of the Syriac bema. The most similar case is represented by Taybbat al Imam (the church already

mentioned for the bema inscription above): it is the same architectonical and decoration project (Figure 13).

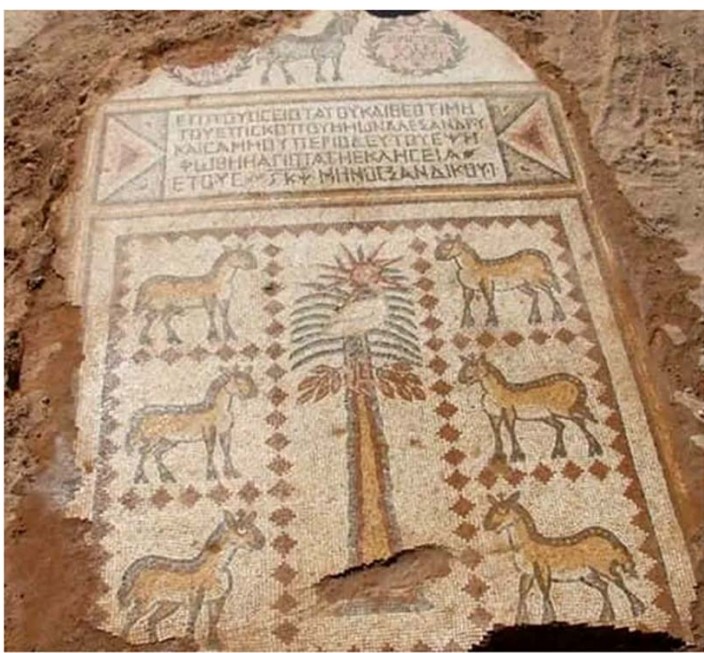

**Figure 11.** Uqayribat: mosaic of the bema (DGAM Syria 2018).

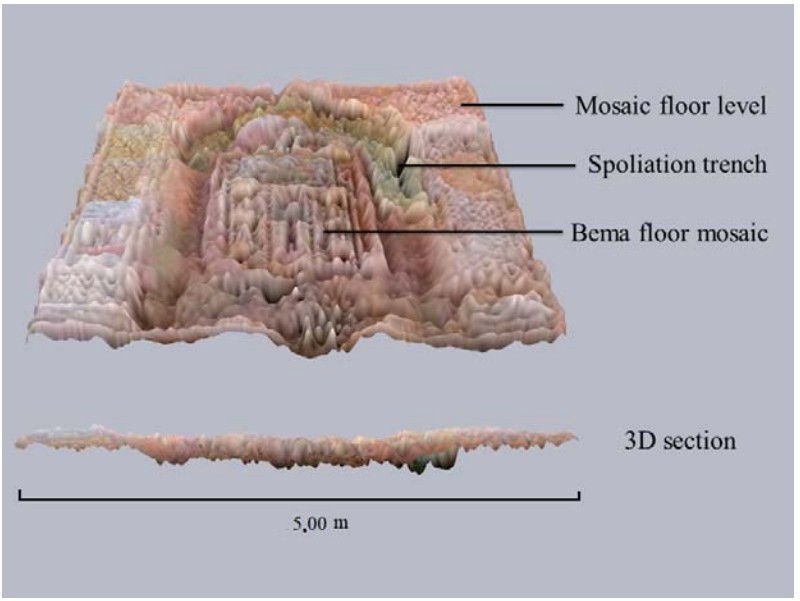

**Figure 12.** Uqayribat—study of the stratigraphic layers: 3D mapping and section evidencing main stratigraphic data (elaboration of Bucci 2020).

The archaeological data from the excavation encourage the thought of a lithic foundation for the U-shaped structure, including the throne. The space inside the U, in both buildings, would have been open to show the mosaic.

The individuation of the stratigraphic main units of the building provides a great contribution to the chronology in relation to the original ancient projects and interior design makeover.

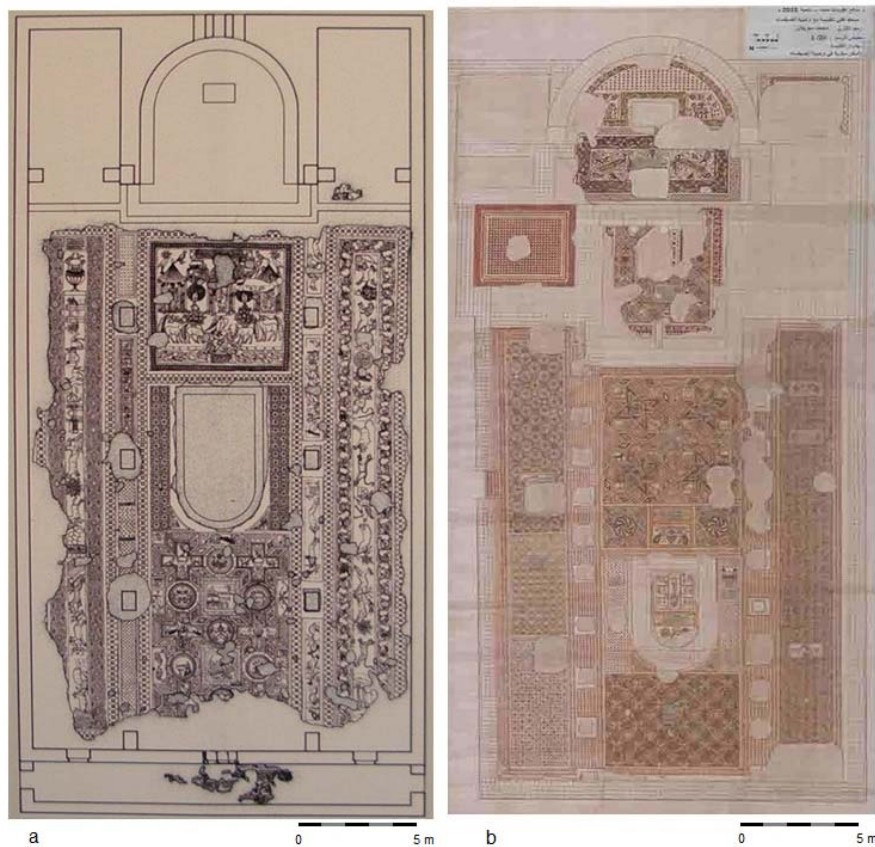

**Figure 13.** Plan views of: (**a**) Taybbat al Imam (Piccirillo, Zaqzuq 1990) and (**b**) Uqayribat (DGAM Syria 2018).

If we consider the example of Dehes, we can conceptualize a synthesis of the construction method and architectonical interconnection. In Dehes, the pavement is still on-site and is situated just against the bema structure; the bema foundation, including the throne, was projected from the original planning of the church; the continuity and homogeneity of the decorations with frames attest to the unitary system of the complex, bema and throne. The foundation of the structure is made of one course of orthostate elements (with semi-curved and rectilinear profiles), partially inserted in a shallow trench (about 20 cm); the second layer of stones comprising the hemicycle rests directly on the first course of blocks, without binder, fixed by gravity The throne, from the static–architectonical point of view, represent a key stone in the middle of the structure: the connection of the stone element with the masonry takes place with a system of interlocking housings, binding the lectern to the sides, gripping the element and ensuring its verticality (Figure 14).

The analysis of the stratigraphic wall units (unit 1: limestone block foundation; unit 2: limestone blocks with support for wooden floor; unit 3: U-shaped limestone bema structure with throne and decoration)—and thanks also to the latitudinal sections of the Syriac bemas detected by Tchalenko in the churches of Bennawi, Qirk Bize, Resafa, Kimar, Kfeir Daret 'Azze, Bafetin, Dehes, Behyo, and Bettir (Figure 15)—confirmed the phases of the lithic lecterns belonging to the original interior design projects of the churches (for more information about rock considerations, please see paragraph five below).

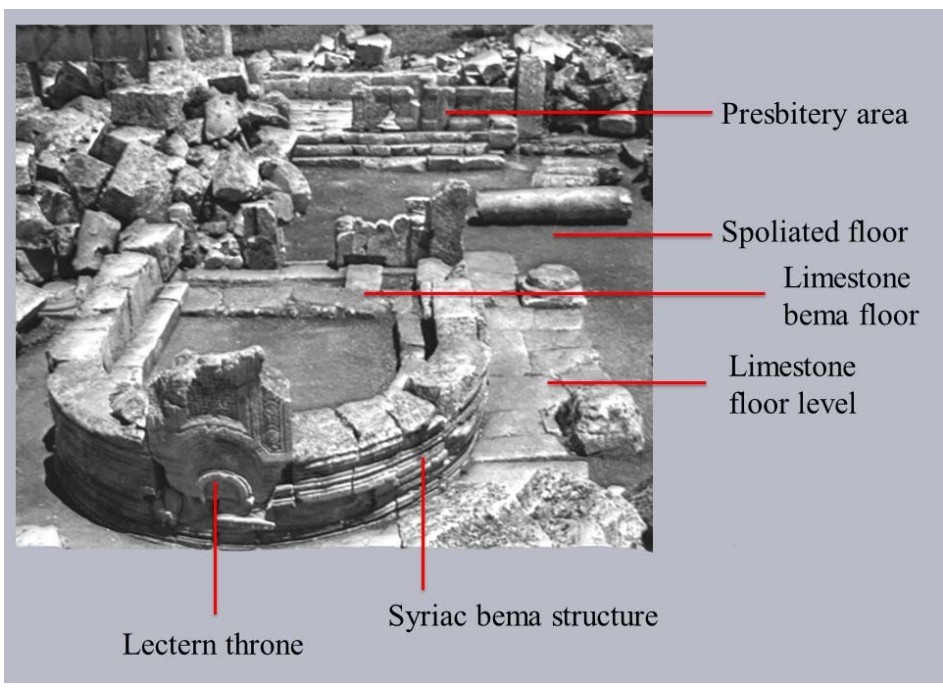

**Figure 14.** Dehes: study of the stratigraphic layers (elaboration from Bucci 2020 of photo from https://medihal.archives-ouvertes.fr/medihal-01849272/thumb).

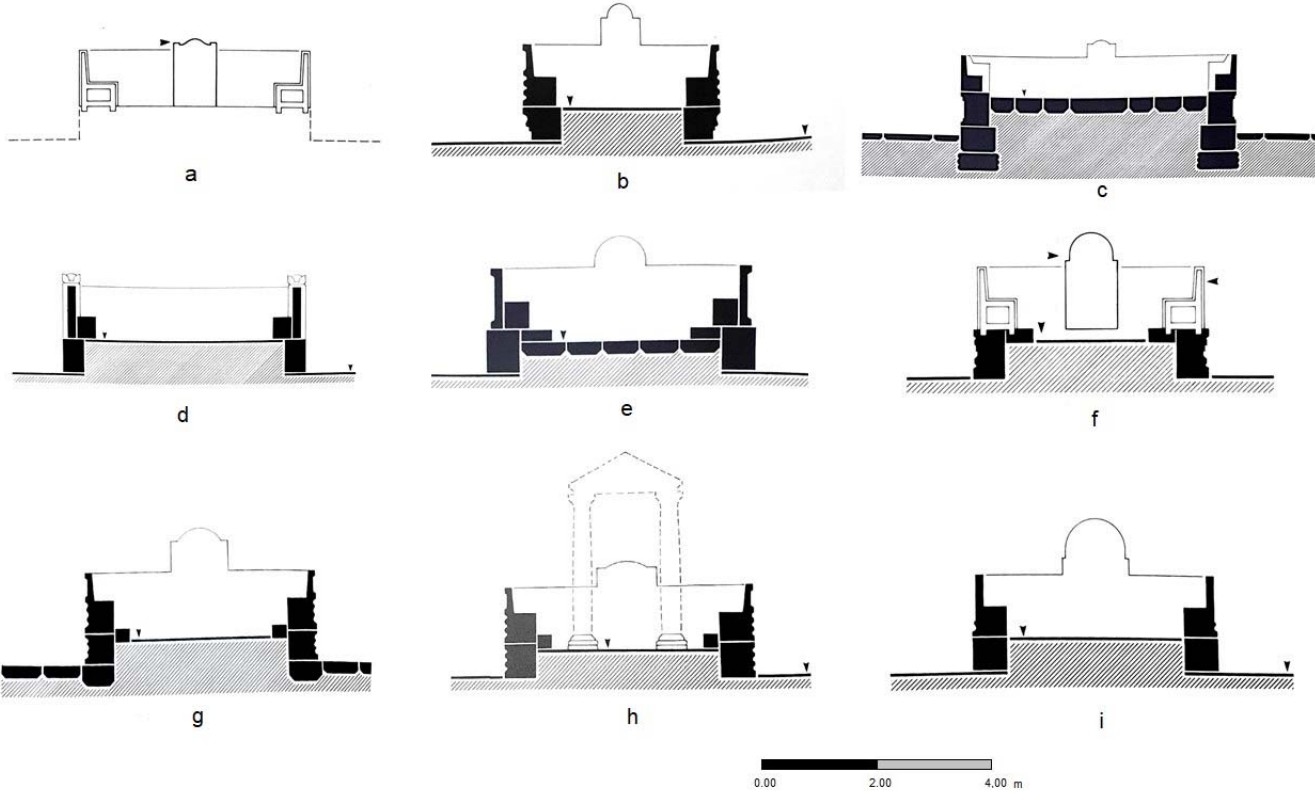

**Figure 15.** Latitudinal sections of Syriac bemas with preserved thrones: (**a**) Bennawi, (**b**) Qirk Bize, (**c**) Resafa, (**d**) Kimar, (**e**) Kfeir Daret 'Azze, (**f**) Bafetin, (**g**) Dehes, (**h**) Behyo, and (**i**) Bettir (elaboration from tables in Tchalenko 1990).

## 6. Mineralogical–Petrographic Considerations

The data presented here came from direct investigations of sites completed between 1994 and 2010 by the author. The typologies of the rocks used for architectural purpose were detected in autoptic surveys and using bibliographical studies (see References). Specific petrographic analyses have not been completed yet (because of the suspension of research activities connected to the civil war still ongoing in Syria).

The building materials for churches and their liturgical furnishings were mainly extracted near the buildings, as mentioned in the introduction; therefore, the structures show homogeneous and monochromatic complexes. These structures were built with materials extracted from open quarries belonging to the surface lithostratigraphy [1–4] (Figure 16).

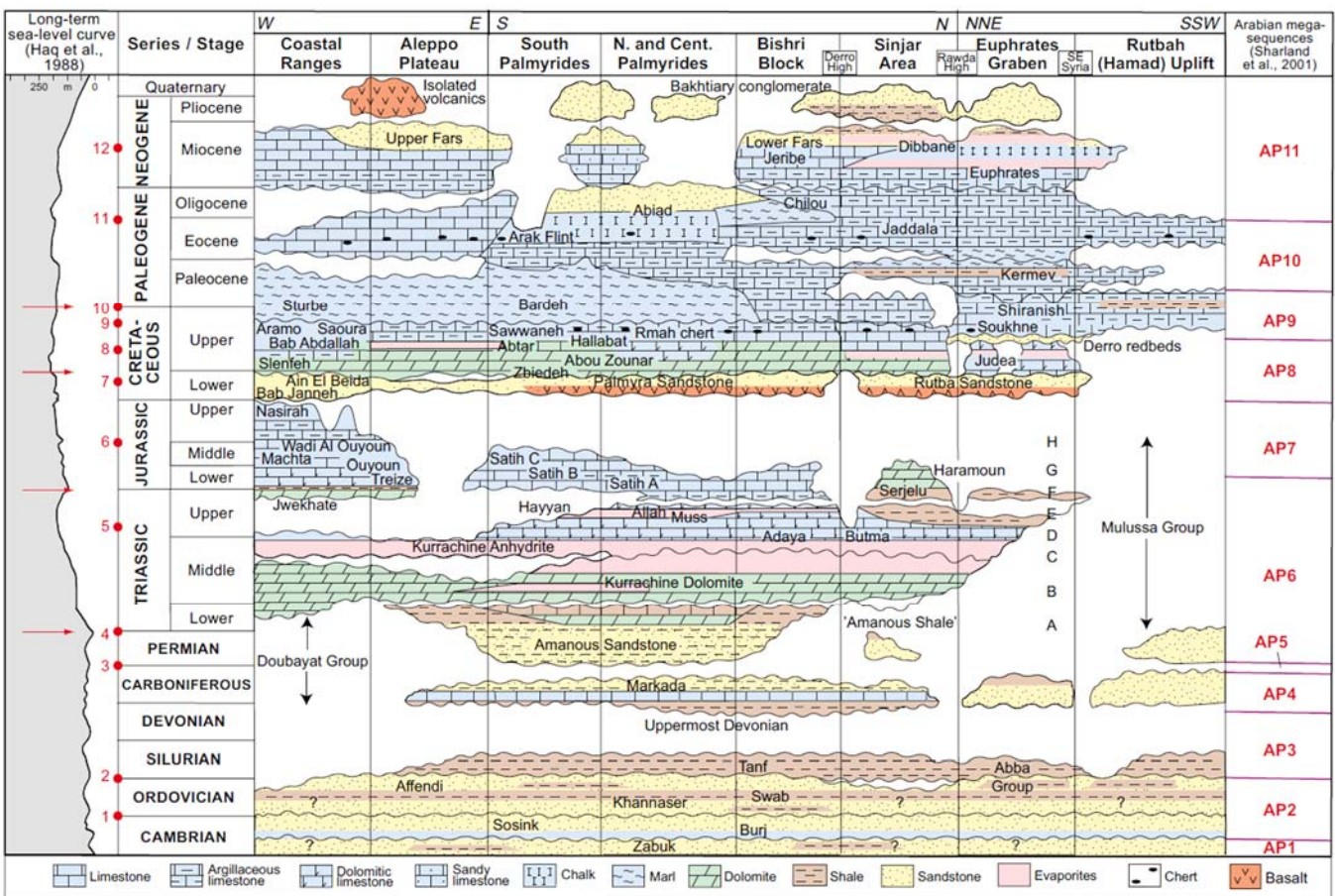

**Figure 16.** Generalized lithostratigraphy of Syria based on surface observations and drilling records (Brew, Barazangi, Al-Maleh, and Sawaf 2001).

The majority of the artifacts here analyzed were made of Syrian limestone, a calcite rock that consists mainly of calcium carbonate ($CaCO_3$), calcite, or aragonite and may also contain a high percentage of magnesium carbonate (dolomite), in addition to some other components in small quantities, such as chloride, iron oxide hydrate, feldspar, and quartz. Limestone, marble, and calcium ($CaCO_3$) are the main component of calcite, which is one of the most widespread raw materials in north and central Syria. $CaCO_3$ is composed of 56% CaO and 44% $CO_2$. It also contains impurities such as $Mg^{+2}$, $Fe^{+2}$, and $Mn^{+2}$ and sometimes $Pb^{+2}$ or $SO4^{-2}$ are crystallized in a cubic crystalline structure. Aragonite consists of 56% CaO and 44% $CO_2$, with some impurities such as $Fe^{2+}$, $Al^{3+}$, $Mg^{2+}$, $Mn^{2+}$, and $SO4^2$. Its crystalline structure differs from that of calcite, as aragonite takes the form of chains of trigonometric carbonate roots ($CO_3^2$), located above each other and associating with $Ca^{2+}$ to form columns extending along the C axis [33].

## 7. Conclusions

The lectern throne or Θρόνος δεύτερος (*thronos deuteros*) is a testament of lithic Christian cultural heritage, an element characterizing the churches of northern Syria during Late Antiquity that demonstrates the use of geological materials in buildings and their furnishings in connection with liturgy. The three typologies here identified show a typical system, attested in a short chronological frame (5th to 6th centuries, with some later modifications).

As indicated by the ancient sources, the lectern throne is a liturgical furnishing device used during liturgy for the βηματικά (bematika), prayers performed when reading holy books, which lay on the lectern. It is a testimony for a religious rite that belonged to the liturgy of Late Antiquity; contextualized as a stone element, with its shape and decoration, it is also a *terminus post quem* useful for the chronological interpretation of the buildings.

**Funding:** This research received no external funding.

**Institutional Review Board Statement:** Not applicable.

**Informed Consent Statement:** Not applicable.

**Data Availability Statement:** Not applicable.

**Conflicts of Interest:** The author declares no conflict of interest.

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
