# Peer review of "Geological Materials in Late Antique Archaeology: The Lithic Lectern Throne of the Christian Syrian Churches"

_heritage, doi:10.3390/heritage4030106_

Round 1

Reviewer 1 Report

Please, find my review of the already revised manuscript No. 1264610: “Geological materials in Late Antique Archaeology: the lithic lectern throne of the Christian Syrian Churches” written by Giovanna Bucci.

The peer-reviewed manuscript has no content that corresponds to the title. Namely, the author puts geology in the foreground in the title, but does not provide basic any original and basic geological data in the text, such as: geological maps, methods used in rock analysis, there are no geological samples from the field and their analysis that should be used to compare with the material from which the studied objects were built, etc.

The author mentions rocks - basalt and limestone, but without description of the applied methods for rock determination. It remains that he tooks the determination of rocks from the literature, in such case used references must be cited!

In addition to this problem, the paper is generally written confusingly. The abstract has only general information. With the exception of geographical terms, almost the entire abstract can be used universally for any manuscript with a similar theme. The introduction lacks data on the names of the sites, on the clearly listed objects that are the subject of the research, on the goals of the research and the methods that have been applied. It is not clear for the whole paper whether it is an original or a reviewed scientific work.

I believe that due to serious and essential shortcomings, this work should be rejected.

Author Response

-

Please, find my review of the already revised manuscript No. 1264610: “Geological materials in Late Antique Archaeology: the lithic lectern throne of the Christian Syrian Churches” written by Giovanna Bucci.

The peer-reviewed manuscript has no content that corresponds to the title. Namely, the author puts geology in the foreground in the title, but does not provide basic any original and basic geological data in the text, such as: geological maps, methods used in rock analysis, there are no geological samples from the field and their analysis that should be used to compare with the material from which the studied objects were built, etc.

The author mentions rocks - basalt and limestone, but without description of the applied methods for rock determination. It remains that he tooks the determination of rocks from the literature, in such case used references must be cited!

In addition to this problem, the paper is generally written confusingly. The abstract has only general information. With the exception of geographical terms, almost the entire abstract can be used universally for any manuscript with a similar theme. The introduction lacks data on the names of the sites, on the clearly listed objects that are the subject of the research, on the goals of the research and the methods that have been applied. It is not clear for the whole paper whether it is an original or a reviewed scientific work.

I believe that due to serious and essential shortcomings, this work should be rejected.

Dear Reviewer 1,

I have added a chapter dedicated to the main geological - petrographic data.

I replyed to the questions, but I think our scientifc approach is very different.

I note, in any case, that my contribution is not of your interest and is probably very far from the research you are involved in. I think that is why it was particularly difficult in the revision.

The other Reviewers are on opposite position and of opposite opinion.

Thanks for your attention

Reviewer 2 Report

I reviewed the manuscript “Geological materials in Late Antique Archaeology: the lithic lectern throne of the Christian Syrian Churches”, by author Giovanna Bucci.

The manuscript is interesting and fits the journal's aims.

The paragraph “philological and epigraphic investigations” provide a good background of the topic as well as the paragraph “The survived exemplars of lectern thrones: description” that is clearly presented.

The motivations for this study are clear and the objective is well defined but considering the title of the manuscript, I was expected more information about the mineralogical/petrographic characterization of the materials considered in this study.

For this reason, the author should improve this part in the manuscript by at least adding some references related to this aspect.

Author Response

Dear Colleague Reviewer 2,
thanks for reviewing my work.
I accepted your suggestion and added a chapter dedicated to the in-depth study on the characterization of materials.
I have also improved some aspects of the images by accepting additional advice from Reviewer 3.
I attach the file with the updates.
Thank you!

Reviewer 3 Report

I find tha the manuscript is interesting and clear. Of course, it is suitable for publication in Heritage. However, the presentation of the work needs some improvements. To improve the manuscript I provide some suggestions:

  • Please, check english as I hace found some gaps.
  • Performe some changes in figures. In the present format they are not clear:

Figure 2. Please, provide a bigger scale. The present one is not visible.

Figure 3. Please, increase the size. Name each sketch with a letter and provide a bigger scale.

Figure 4. Please, provide a bigger scale. The present one is not visible. 

Figure 8. Please, provide a scale.

Figure 12. Please, provide a scale.

Figure 13. Please, provide a scale.

Figure 15. Please, increase the size. Name each sketch with a letter and provide a bigger scale. 

Author Response

Dear Colleague Reviewer 3,
grants for reviewing my work.
I accepted all the suggestions related to the figures, corrected some oversights and added a specific chapter on the petrographic characterization of materials.
I am attaching the updated file with corrections.
Thank you!
